# Phase I/II Clinical Trial of the Anti-Podoplanin Monoclonal Antibody Therapy in Dogs with Malignant Melanoma

**DOI:** 10.3390/cells9112529

**Published:** 2020-11-23

**Authors:** Satoshi Kamoto, Masahiro Shinada, Daiki Kato, Sho Yoshimoto, Namiko Ikeda, Masaya Tsuboi, Ryohei Yoshitake, Shotaro Eto, Yuko Hashimoto, Yosuke Takahashi, James Chambers, Kazuyuki Uchida, Mika K. Kaneko, Naoki Fujita, Ryohei Nishimura, Yukinari Kato, Takayuki Nakagawa

**Affiliations:** 1Laboratory of Veterinary Surgery, Graduate School of Agricultural and Life Sciences, The University of Tokyo, 1-1-1, Yayoi, Bunkyo-ku, Tokyo 113-8657, Japan; veteronut@gmail.com (S.K.); ms003_shina@yahoo.co.jp (M.S.); syoshimo@g.ecc.u-tokyo.ac.jp (S.Y.); tsubaki5228@gmail.com (N.I.); g050446@gmail.com (R.Y.); shotaro.nyantaro@gmail.com (S.E.); apom@g.ecc.u-tokyo.ac.jp (N.F.); surgspartan@g.ecc.u-tokyo.ac.jp (R.N.); anakaga@g.ecc.u-tokyo.ac.jp (T.N.); 2Veterinary Medical Center, The University of Tokyo, 1-1-1, Yayoi, Bunkyo-ku, Tokyo 113-8657, Japan; atsuboi@g.ecc.u-tokyo.ac.jp (M.T.); yuko14hashi@gmail.com (Y.H.); yousuke1982_root_story@yahoo.co.jp (Y.T.); 3Laboratory of Veterinary Pathology, Graduate School of Agricultural and Life Sciences, The University of Tokyo, 1-1-1, Yayoi, Bunkyo-ku, Tokyo 113-8657, Japan; achamber@g.ecc.u-tokyo.ac.jp (J.C.); auchidak@g.ecc.u-tokyo.ac.jp (K.U.); 4Department of Antibody Drug Development, Tohoku University Graduate School of Medicine, 2-1, Seiryo-machi, Aoba-ku, Sendai 980-8575, Japan; k.mika@med.tohoku.ac.jp (M.K.K.); yukinarikato@med.tohoku.ac.jp (Y.K.); 5New Industry Creation Hatchery Center, Tohoku University, 2-1, Seiryo-machi, Aoba-ku, Sendai 980-8575, Japan

**Keywords:** podoplanin, dog, antibody therapy, tumor, melanoma

## Abstract

Podoplanin (PDPN), a small transmembrane mucin-like glycoprotein, is ectopically expressed on tumor cells. PDPN is known to be linked with several aspects of tumor malignancies in certain types of human and canine tumors. Therefore, it is considered to be a novel therapeutic target. Monoclonal antibodies targeting PDPN expressed in human tumor cells showed obvious anti-tumor effects in preclinical studies using mouse models. Previously, we generated a cancer-specific mouse–dog chimeric anti-PDPN antibody, P38Bf, which specifically recognizes PDPN expressed in canine tumor cells. In this study, we investigated the safety and anti-tumor effects of P38Bf in preclinical and clinical trials. P38Bf showed dose-dependent antibody-dependent cellular cytotoxicity against canine malignant melanoma cells. In a preclinical trial with one healthy dog, P38Bf administration did not induce adverse effects over approximately 2 months. In phase I/II clinical trials of three dogs with malignant melanoma, one dog vomited, and all dogs had increased serum levels of C-reactive protein, although all adverse effects were grade 1 or 2. Severe adverse effects leading to withdrawal of the clinical trial were not observed. Furthermore, one dog had stable disease with P38Bf injections. This is the first reported clinical trial of anti-PDPN antibody therapy using spontaneously occurring canine tumor models.

## 1. Introduction

Podoplanin (PDPN), also known as PA2.26, gp38, T1α, and Aggrus, is a type I transmembrane sialoglycoprotein expressed in various types of tissues, including renal podocytes, pulmonary type I alveolar cells, and lymphatic endothelial cells [1,2,3,4]. PDPN plays an essential role in the development of the lymphatic system in the embryo and in platelet aggregation [1,2]. In human medicine, PDPN has been reported to be overexpressed in various types of tumors, including squamous cell carcinoma [5], astrocytoma [6], malignant mesothelioma [7], hemangiosarcoma [8], osteosarcoma [9], germinoma [10], and cancer-associated fibroblasts (CAFs) [11,12,13]. Similar to humans, PDPN has been reported to be expressed in renal podocytes, alveolar epithelial cells, and lymphatic endothelial cells of dogs, and in various types of canine tumors, including malignant melanoma and squamous cell carcinoma [14,15,16,17,18]. Many reports have demonstrated that PDPN expressed on human and canine tumors is associated with tumor malignancy through the promotion of malignant proliferation and epithelial–mesenchymal transition (EMT), and that it promotes metastasis by enhancing tumor cell migration and platelet aggregation [1,2,18,19,20,21]. These reports promoted the further evaluation of PDPN as a therapeutic target.

Immunotherapy using therapeutic antibodies, which have blocking activity between a receptor and ligand, or antibody-dependent cellular cytotoxic (ADCC) activity, have become the predominant class of new drugs developed in recent years for various tumors [22,23,24,25,26,27]. Recently, PDPN has attracted attention as a novel target antigen for the development of antibody therapy because PDPN is expressed on various refractory tumors. As a blocking antibody, the anti-PDPN neutralizing antibodies MS-1 and SZ168, which inhibit the binding of PDPN expressed on tumor cells and C-type lectin-like receptor 2 expressed on platelets, decreased tumor growth and metastasis in PDPN-overexpressed Chinese hamster ovary (CHO)-K1 cells and human melanoma cell lines xenografted onto mouse models [28,29]. Furthermore, the anti-human PDPN antibody, NZ-12, showed an obvious anti-tumor effect against human malignant pleural mesothelioma in an orthotopic xenograft model by inducing ADCC activity [30]. However, it is expected that anti-PDPN cytotoxic antibodies might cause adverse effects because the antibodies could bind to both PDPN-expressing tumors and normal tissues. To solve this issue, we have established a cancer-specific monoclonal antibody (CasMab) against human PDPN, which recognizes the cancer-specific aberrant glycosylation of human PDPN [31]. By using anti-PDPN CasMab, it is possible to specifically target PDPN expressed on tumor tissues but not on normal tissues. Furthermore, anti-PDPN CasMab inhibited the growth and pulmonary metastasis of human-PDPN-expressing tumors in vivo [32]. Based on these strategies, immunotherapy using a therapeutic antibody targeting PDPN expressed on tumor cells is considered to be a promising therapy for patients with PDPN-positive tumors. However, an anti-tumor effect of anti-PDPN antibody therapy has so far been proven only in mouse models, and evaluation of the safety of PDPN-targeting therapies is required before first-in-human clinical trials.

Dogs with tumors are often used as research models of human patients with tumors because dogs live in close proximity to humans. They are influenced by similar environmental factors that can lead to cancer development, and many features of spontaneously occurring canine tumors are similar to those of human tumors in terms of histological morphology, biological behavior, molecular mechanisms, and response to conventional therapy [33,34,35]. To perform clinical trials using spontaneously occurring canine tumor models, we developed a cancer-specific anti-canine PDPN (dPDPN) monoclonal antibody (mAb), PMab-38, which recognizes only dPDPN expression in canine tumor tissues without recognizing normal canine tissues [16]. Immunohistochemical evaluation demonstrated 90% of canine melanoma, 83% of canine squamous cell carcinoma, and 82% of canine pulmonary adenocarcinoma cells, and CAFs were positively stained with PMab-38, while normal canine tissues were hardly stained by PMab-38 [17,18]. Furthermore, a cancer-specific mouse–dog chimeric anti-PDPN antibody (P38Bf) originating from mouse monoclonal PMab-38 were shown to reduce immunological reactions, and P38Bf demonstrated strong anti-tumor effects in a PDPN-overexpressed CHO-K1 cell xenograft mouse model [36,37].

The objective of this study was to evaluate the efficacy and safety of a cancer-specific anti-dPDPN chimeric antibody: P38Bf. The ADCC activity of the antibody against endogenously dPDPN-expressing canine tumor cells was evaluated in vitro. The safety and efficacy of the antibody in clinical cases was evaluated in a phase I/II clinical trial.

## 2. Materials and Methods

### 2.1. Cell Culture

Five canine melanoma cell lines (CMM1, CMM2, CMM8, CMM11, and KMeC) were used [38,39,40,41]. CMM1, CMM2, and KMeC were maintained in RPMI-1640 medium (FUJIFILM Wako Pure Chemical Corporation, Osaka, Japan) supplemented with 10% fetal bovine serum (FBS; Cosmo Bio Co., Tokyo, Japan) and 50 mg/L gentamicin sulfate (Sigma Chemical Co., St Louis, MO, USA). CMM8 and CMM11 were maintained in D-MEM/Ham’s F-12 medium (FUJIFILM Wako Pure Chemical Corporation) supplemented with 10% FBS, 100 U/mL penicillin, 100 μg/mL streptomycin, and 250 ng/mL amphotericin B suspension (PSA; FUJIFILM Wako Pure Chemical Corporation). Chinese hamster ovary (CHO)-K1 and canine PDPN (dPDPN)-expressing CHO-K1 with N-terminal PA tag (CHO/dPDPN) cell lines were also used [15]. CHO-K1 and CHO/dPDPN were cultured in RPMI-1640 medium supplemented with 10% FBS and PSA. All cell lines were incubated at 37 °C in a humid atmosphere with 5% CO_2_. These conditions were used in the following experiments, unless otherwise stated.

### 2.2. Antibody

PMab-38, a mouse anti-dPDPN mAb (ImmunoglobulinG_1_ (IgG_1_), kappa), was developed as previously described [15]. To generate P38B, a mouse-canine (subclass B) chimeric antibody, the appropriate V_H_ and V_L_ cDNAs of mouse PMab-38 and the C_H_ and C_L_ of canine IgG subclass B were subcloned into pCAG-Ble and pCAG-Neo vectors (FUJIFILM Wako Pure Chemical Corporation) [36]. To generate a cancer-specific mouse–dog chimeric anti-PDPN antibody (P38Bf), antibody expression vectors were transfected into BINDS-09 (FUT8-knocked out ExpiCHO-S cells; http://www.med-tohoku-antibody.com/topics/001_paper_cell.htm) using the ExpiCHO-S Expression System (Thermo Fisher Scientific, Waltham, MA, USA). P38Bf was purified using Protein G-Sepharose (GE Healthcare Bio-Sciences, Pittsburgh, PA, USA).

### 2.3. Enzyme-Linked Immunosorbent Assay (ELISA)

The concentration of P38Bf in canine serum was measured by a direct ELISA that assayed for binding to canine PDPN. Canine PDPN protein was prepared from the detergent-soluble membrane fraction of canine CHO/dPDPN cells, as previously described [15]. After solubilization, we used the PA tag system for the purification of canine PDPN from cell extracts [42]. Purified proteins were immobilized on Nunc Maxisorp 96-well immunoplates (Thermo Fisher Scientific) at 1 μg/mL for 30 min. After blocking with SuperBlock T20 Blocking Buffer (Thermo Fisher Scientific), the plates were incubated with serially diluted serum samples followed by 1:5000 diluted peroxidase-conjugated anti-dog IgG (Thermo Fisher Scientific). Linear standard curves of P38Bf were generated from 3.9 to 125 ng/mL. The enzymatic reaction was produced with a 1-Step Ultra 3,3′,5,5′-tetramethylbenzidine-ELISA (Thermo Fisher Scientific). The optical density was measured at 655 nm using an iMark microplate reader (Bio-Rad Laboratories, Hercules, CA, USA). All measurements were performed in triplicate, and each reaction was conducted at 37 °C with a total sample volume of 50 μL.

### 2.4. Flow Cytometry 

Cells were harvested after a brief exposure to 0.25% trypsin/1 mM ethylenediaminetetraacetic acid (EDTA; Nacalai Tesque, Inc., Kyoto, Japan). After 3 washes in fluorescence-activated cell sorting (FACS) buffer (phosphate buffered saline (PBS) containing 5% FBS and 0.1% sodium azide), cells were incubated with 5 μg/mL mouse-canine chimeric anti-dPDPN antibody (P38Bf) for 30 min at 4 °C, followed by 3 washes in FACS buffer and incubation with a 1:800 dilution of Alexa Fluor^®^ 647-conjugated anti-dog IgG antibodies (Jackson ImmunoResearch Inc., West Grove, PA, USA) for 30 min at 4 °C. Whole molecule dog IgG (Jackson ImmunoResearch) was used as an isotype control. All flow cytometric analyses were performed with BD FACSverse (BD, Franklin Lakes, NJ, USA) and data were analyzed using BD FACSuite software (BD, ver. 8.0).

### 2.5. Evaluation of Antibody-Dependent Cellular Cytotoxicity (ADCC)

ADCC was examined using a calcein-acetyoxymethyl (Calcein-AM; Cayman Chemical Co., Ann Arbor, MI, USA) release assay. Canine lymphokine-activated killer (LAK) cells were prepared by culturing canine peripheral blood mononuclear cells from a healthy beagle dog in the presence of 1000 IU/mL of human recombinant interleukin-2 (IL-2) (Novartis, East Hanover, NJ, USA) for 1 week, as previously reported [36]. CMM2 and KMeC cells were used as target cells. The target cells were labeled with Calcein-AM for 30 min, washed 3 times with PBS containing 5% FBS, and plated onto 96-well plates at a density of 1 × 10^4^ cells/well. P38Bf or whole molecule dog IgG was added at various concentrations from 0.01 to 10 μg/mL for 15 min on ice. Next, the LAK cells were added as effector cells at an effector (E)/target (T) ratio of 10:1. Then, the plates were incubated for 4 h at 37 °C, and the relative light units (RLU) of the supernatants were analyzed using fluorometry to measure calcein release (cell death). For maximal release, the cells were lysed with 2% Triton X-100. Fluorescence was detected using the ARVO X4 system (PerkinElmer, Waltham, MA, USA). ADCC activity was calculated using the following formula:Cytotoxicity (%) = (test RLU − spontaneous death RLU)/(maximal deathRLU − spontaneous death RLU) × 100.

For each experiment, measurements were conducted in quadruplicate using four replicate wells. Each experiment was repeated at least 3 times.

### 2.6. P38Bf Injection into a Healthy Dog

To examine the safety of anti-dPDPN treatment in dogs, P38Bf (2 mg/kg) was intravenously administrated into one healthy beagle (female, 1.2 years old) once every 2 weeks for a total of four administrations. The study was approved by the University of Tokyo Animal Care and Use Committee (P18-021). A thorough physical examination, complete blood count (CBC) (IDEXX ProCyte Dx, IDEXX Lab., ME, USA), serum chemistry (albumin (ALB), alkaline phosphatase (ALP), blood urea nitrogen (BUN), calcium (Ca), creatinine (CRE), C-reactive protein (CRP), electrolytes (Na, K, Cl), glomerular filtration rate (GFR), γ-glutamyltransferase (GGT), glucose (GLU), glutamic oxaloacetic transaminase; glutamic-oxaloacetic transaminase (GOT), glutamic pyruvic transaminase (GPT), total bilirubin (TBIL), total cholesterol (TCHO), triglycerides (TG), total protein (TP), lipase (vLIP), and inorganic phosphorus (iP)) (Fuji DRI-CHEM 7000V, FUJIFILM Co., Tokyo, Japan), and urinalysis (Thinka Urine Test Strip and Urine Analyzer Thinka RT-4010, Arkray Co., Kyoto, Japan) were performed to determine general health status. To evaluate the acute adverse effects, body temperature, heart rate, respiratory rate, and percutaneous oxygen saturation (SpO_2_) were recorded at baseline and every 30 min after the administration of P38Bf for 5 h. SpO_2_ was monitored using a bedside monitor (LifeScope, NIHON KOHDEN Co., Tokyo, Japan). To evaluate chronic adverse effects, blood tests were repeated every day for 7 days following antibody administration and repeated every week thereafter. Body temperature, heart rate, and respiratory rate were monitored at the same time as the blood test. Since PDPN is expressed in the kidney glomerulus, plasma clearance of intravenously administered inulin was evaluated for estimating the glomerular filtration rate to obtain detailed information on kidney function (FUJIFILM Monolith Co). Computed tomography (CT) was performed to evaluate structural abnormalities in the whole body at the timing of, before 7 days, and after 56 days of antibody administration. Before and after injection of the antibody, concentrations of serum P38Bf were measured by direct ELISA, as mentioned above. The dog was sacrificed 2 weeks after the last administration for gross necropsy and histopathological analysis. The pathological evaluation was performed by a single veterinary pathologist. Adverse events were assessed and classified according to the Veterinary Cooperative Oncology Group—Common Terminology Criteria for Adverse Events (VCOG-CTCAE) v 1.1 criteria [43].

### 2.7. Immunohistochemistry

Immunohistochemical staining was performed using primary antibodies specific for dPDPN to determine the eligibility for enrollment in the clinical trial of P38Bf treatment. All tumor tissues were fixed in 10% neutral buffered formalin, embedded in paraffin wax, and cut into 4 µm serial sections. Paraffin-embedded tumor sections were dewaxed and rehydrated in xylene and graded ethanol, followed by antigen retrieval using 10 mM Tris-HCl 1 mM EDTA buffer pH 9.0 at 100 °C for 30 min in boiling water. After washing with Tris-buffered saline with 0.1% Tween^®^ 20 (Sigma Chemical Co) detergent (TBST), endogenous peroxidase was blocked with 3% H_2_O_2_ in methanol for 10 min at room temperature. Then, specimens were washed with TBST and incubated in 8% skim milk for 1 h at room temperature to reduce nonspecific binding before overnight incubation with primary antibodies, including mouse IgG1 anti-dPDPN mAb (PMab-38) diluted 1:200 at 4 °C in a humidified chamber. A negative control was incubated with the purified mouse IgG1 κ isotype antibody (Clone: MG1-45, BioLegend, San Diego, CA, USA) under identical conditions. After washing with TBST, sections were incubated with a horseradish peroxidase-conjugated anti-mouse antibody (EnVision™+ System, a horseradish peroxidase (HRP) labeled polymer; K4001; Agilent Technologies, Santa Clara, CA, USA) for 30 min at room temperature. Thereafter, the sections were washed with TBST, incubated with 3,3′ diaminobenzidine (Dojindo Laboratories, Kumamoto, Japan) solution for 3 min, and counterstained with Mayer’s hematoxylin. ALP staining was performed in melanoma tissues with numerous melanin granules. In this method, ALP-conjugated streptavidin (diluted to 1:400; Innova Biosciences, Cambridge, UK) and an ALP substrate kit (VECTOR Laboratories, Burlingame, CA, USA) were used in place of HRP-conjugated streptavidin and 3,3′-diaminobenzidine. Sections of canine squamous cell carcinoma tissues were used as positive controls for dPDPN. The specimens were considered positive for dPDPN if histological evidence of cell staining was present in five independent high-power (400× magnification) fields.

### 2.8. Clinical Trial in Dogs with Malignant Tumors

To evaluate the clinical safety and efficacy of P38Bf, a clinical trial was conducted at the Veterinary Medical Center (VMC), University of Tokyo. The study was approved by the Animal Care and Clinical Research Committees of the VMC, University of Tokyo (VMC2018-4). The inclusion criteria for the clinical trial were dogs with dPDPN positive malignant melanoma and tumors resistant to standard therapies (e.g., surgery, radiation therapy, and chemotherapy). dPDPN expression was confirmed by immunohistochemical analysis of the tumor tissues obtained by surgical excision prior to surgery or biopsy. After written informed consent was obtained from the owners, P38Bf was intravenously administered every 2 weeks at 2 mg/kg, at a rate of 1 mL/min or below, except for the indicated case. During the treatment period, the dogs were monitored by physical examination, CBC, and serum chemistry at least every 2 weeks. The monitoring time course was the same as that of the safety evaluation using a healthy dog (2.4.). The tumor size was measured using a caliper and recorded every 2 weeks if measurable lesions were present on the body surface. At baseline (within 2 weeks prior to the first P38Bf administration), the day of the third administration, and 2 weeks after the fourth administration, thoracic radiography or CT were performed to evaluate the tumor burden in the whole body. The tumor burden was calculated as the sum of the longest diameters of all measurable target lesions. Tumors ≥ 10 mm through the longest diameter were considered measurable lesions. A maximum of five target lesions were chosen from measurable lesions at baseline, with a maximum of two lesions per organ. Tumor response to P38Bf treatment was defined as follows: complete response (CR) was disappearance of all detectable tumor; partial response (PR) was at least 30% reduction in the sum of the diameters of target lesions; stable disease (SD) was less than 20% increase or 30% reduction in the sum of diameters, and progressive disease (PD) was at least a 20% increase in the sum of diameters. The longest diameters of new measurable lesions (up to five lesions in total and up to two lesions per organ) were included in the sum. The clinical responses were evaluated for each administration of P38Bf according to response evaluation criteria for solid tumors (RECIST) in dogs (v1.0) [44]). Adverse events were assessed and classified according to the VCOG-CTCAE v 1.1 criteria [43]. World Health Organization staging was used for staging.

### 2.9. Statistics

All data are shown as the mean ± standard error (SE). For the evaluation of ADCC activity, Tukey–Kramer’s test was performed using R software (ver. 3.6.1, R Development Core Team, 2019). For the evaluation of acute and chronic adverse effects, a one-way ANOVA test was performed using R software. Values of *p* < 0.05 were considered statistically significant.

## 3. Results

### 3.1. Antibody-Dependent Cellular Cytotoxicity (ADCC) Induced by a Cancer-Specific Mouse–Dog Chimeric anti-PDPN Antibody (P38Bf) against Canine Melanoma Cells

The reactivity of P38Bf against canine podoplanin (dPDPN) expression on the cell surface of canine melanoma cell lines was evaluated using flow cytometric analysis. Chinese hamster ovary (CHO)-K1 and canine PDPN-expressing CHO-K1 with N-terminal PA tag (CHO/dPDPN) cells were used as negative and positive controls, respectively. dPDPN expression on CHO/dPDPN, CMM2, and CMM11 cells were clearly detected by P38Bf (Figure 1A and Appendix A). CHO-K1 cells showed no fluorescence, and KMeC, CMM1, and CMM8 showed low fluorescence intensity (Figure 1B and Appendix A).

From the results of the flow cytometric analysis, CMM2 and KMeC were selected as dPDPN-high and dPDPN-low expressing cells, respectively. Using CMM2 and KMeC cells, the ADCC activity induced by P38Bf was evaluated. P38Bf showed strong cytotoxicity against CMM2 at antibody concentrations of 1 and 10 µg/mL compared to the control isotype antibody (*p* < 0.01, Figure 1C). In contrast, no significant ADCC activity of P38Bf was measured in KMec cells (Figure 1D).

### 3.2. Safety and Toxicity of P38Bf

As P38Bf showed significant in vitro anti-tumor activity, we next evaluated the detailed adverse effects of P38Bf in an experimental healthy beagle dog. P38Bf (2.0 mg/dog) was intravenously injected into a healthy beagle dog once every 2 weeks for a total of four administrations. The maximum plasma concentration of P38Bf was 23.09 µg/mL and P38Bf was detected even after 168 h (4.72 µg/mL) of administration (Appendix A). After 336 h of P38Bf administration, P38Bf was no longer detectable in the plasma (below detection threshold of 3.9 µg/mL).

Acute adverse effects were evaluated at each administration (4 times) for 5 h after P38Bf administration. Body temperature (mean: 38.6 °C, range: 37.8–39.3 °C), heart rate (mean: 96/min, range: 84–108/min), respiratory rate (mean: 18/min, range: 12–24/min), and percutaneous oxygen saturation (SpO_2_) (mean: 99%, range: 97–100%) were within normal range during the 5 h period (Appendix A). Chronic adverse effects were evaluated for 8 weeks after the P38Bf administration by physical tests (body temperature, heart rate, respiratory rate, and SpO_2_) and blood tests (complete blood count (CBC), albumin (ALB), alkaline phosphatase (ALP), blood urea nitrogen (BUN), calcium (Ca), creatinine (CRE), C-reactive protein (CRP), electrolytes (Na, K, Cl), glomerular filtration rate (GFR), γ-glutamyltransferase (GGT), glucose (GLU), glutamic oxaloacetic transaminase; glutamic-oxaloacetic transaminase (GOT), glutamic pyruvic transaminase (GPT)), total bilirubin (TBIL), total cholesterol TCHO), triglycerides (TG), total protein (TP), lipase (vLIP), inorganic phosphorus (iP)). All evaluated parameters were within the normal range during the observation period (Appendix A and Appendix A). Body weight did not change substantially (mean: 9.8 kg, range: 9.4–10.3 kg) (Appendix A). No abnormal changes were found by urinalysis (data not shown). After 56 days of antibody administration, computed tomography (CT) examination was performed to evaluate organic changes by P38Bf between pre-first administration and post-last administration, and there was no detectable change including inflammation or necrosis in systemic organs (Movie S1 (pre-first) and S2 (post-last)). After 8 weeks of administration, the dog was euthanized, and adverse histological effects in each organ were evaluated, including kidney, lung, small intestine, stomach, bladder, spleen, bone marrow, and lymph node. No adverse events or abnormalities in histological findings were found in the systemic organs (Appendix A).

Following the data of the preclinical study using the experimental healthy dog, we conducted a phase I/II clinical trial for the evaluation of clinical safety and potential efficacy of anti-PDPN antibody therapy in dogs with dPDPN-positive malignant melanoma. Three dogs with malignant melanoma that diffusely expressed dPDPN were enrolled in the clinical trial, and the clinical characteristics of each dog are shown in Table 1. The representative images of dPDPN expression in each tumor tissue are shown in Appendix A. Breeds of included dogs were mixed breed, miniature dachshund, and beagle, respectively, and their ages ranged from 13.7 to 14.8 years (median: 14.7 years) at the time of enrollment. All dogs were resistant to conventional therapies including surgery, chemotherapy, and radiation therapy. Detailed clinical histories of each case are summarized in Table 1. The dogs were intravenously injected with P38Bf at 2 mg/kg. Dog#2 received 4 injections of P38Bf with a two-week interval, except for the fourth administration, which was performed 3 weeks after the third administration for reasons pertaining to the owner. Dog#1 and #3 received a single administration.

For the evaluation of acute adverse effects, changes in body temperature, heart rate, and respiratory rate were monitored for up to 3 h after the intravenous administration of P38Bf. Body temperature (mean: 38.3 °C, range: 37.6–39.4 °C), heart rate (mean: 118/min, range: 57–152/min), and respiratory rate (mean: 21/min, range: 16–32/min) were within the normal range during the 3 h period, and there was no significant change (Figure 2A–C). Since grade 2 vomiting was observed in dog#2 within 3 h of administration, maropitant (1 mg/kg, s.c.) was administered as an anti-emetic drug (Table 2). Vomiting could be controlled by a single anti-emetic drug administration, and treatment was not withdrawn. On the day after the first administration, the observed treatment-related adverse events were all grade 1 or 2 and did not cause treatment withdrawal. During the observation period, chronic adverse effects were evaluated, including the monitoring of symptoms, physical tests, and blood tests. The evaluated parameters showed no significant change or abnormality except for CRP increase in the blood chemistry tests of all dogs (Table 2 and Appendix A). All dogs showed mild CRP increases without any symptoms. The maximum of the CRP level of dog#1 was 7.0 pg/mL. The maximum of the CRP level of dog#2 was also 7.0 pg/mL at 70 days after antibody administration. After 70 days, the CRP level gradually decreased, and the CRP level was 4.2 pg/mL at 84 days after antibody administration. The maximum of the CRP level of dog#3 was also 7.0 pg/mL after 8 days of antibody administration, and the CRP level was 0.8 pg/mL at 14 days after antibody administration.

### 3.3. Clinical Response of P38Bf

The overall responses were stable disease (SD) in one dog (#3) and progressive disease (PD) in two dogs (#1 and #2) (Table 3).

Dog#1 was categorized as WHO stage IV and had metastatic lesions in the lung and skin, which expanded rapidly even with some conventional therapies before inclusion in this clinical trial. The primary tumor in the oral cavity was a microscopic lesion after palliative surgery and was not measurable during the evaluated period. Dog#1 showed a new metastatic tumor lesion in the skin after 7 days of antibody administration and was diagnosed with PD. Four lesions of skin metastasis existed before inclusion in the study. Tumor growth of one lesion in skin was stopped, and one lesion in skin was decreased after P38Bf administration (Figure 3A). Although part of the lung metastasis showed growth inhibition, the majority of the metastatic lesions of the lung tended to develop even after P38Bf administration (Appendix A).

Dog#2 was categorized as WHO stage III and had a 31.1 mm primary tumor in the oral cavity without metastatic lesions. The long axis diameter of the primary tumor in the oral cavity was measured using a CT exam at the day of antibody administration and after 49 days of antibody administration. Dog#2 received antibody administration four times. Although the metastatic lesion was not detected during the observation period, 49 days after first administration, the tumor diameter had increased by about 40% compared to the time of first administration (Figure 3B and Appendix A). Thus, dog#2 was categorized as PD.

Dog#3 was categorized as WHO stage I and had a 13.5 mm primary tumor in the oral cavity without metastatic lesions. The long diameter of the primary tumor in the oral cavity was measured. Dog#3 was categorized as SD because the tumor burden of the primary tumor increased by less than 20%, and new metastatic lesions were not detected during the observation period (Figure 3C). The long diameter increased by only 2% and 8% at 2 and 4 weeks after P38Bf administration, respectively, compared to the time of first administration, although it increased by 14% during the 2 weeks before P38Bf administration.

## 4. Discussion

This is the first report of a clinical trial using anti-podoplanin (PDPN) mAb in canine models. We first demonstrated that a cancer-specific mouse–dog chimeric anti-PDPN antibody (P38Bf) induces antibody-dependent cellular cytotoxicity (ADCC) by recognizing endogenously expressed canine podoplanin (dPDPN) on canine malignant melanoma cells. In this clinical trial, we evaluated the acute and chronic adverse effects of P38Bf in one healthy experimental dog and three dogs with canine malignant melanoma, and we did not find any serious adverse effects during the experimental periods. Furthermore, P38Bf might have delayed tumor growth in several dogs.

In this study, the reactivity and ADCC of P38Bf were evaluated using canine malignant melanoma cell lines. This study targets canine malignant melanoma because it was reported that 80% of canine malignant melanomas express dPDPN on tumor cells [18,45]. P38Bf clearly recognized endogenously expressed dPDPN on canine malignant melanoma cell lines and showed significant and dose-dependent ADCC activity on CMM2 at antibody concentrations of 1 and 10 µg/mL. The specificity of ADCC activity against dPDPN expression induced by P38Bf was originally demonstrated using anine PDPN-expressing chinese hamster ovary (CHO)-K1 with N-terminal PA tag (CHO/dPDPN) cells (dPDPN overexpressing cells) and CHO-K1 cells (dPDPN negative cells) [36], and therefore, these cell lines were included in the current study as positive and negative controls. [36]. Therefore, it is suggested that an induction of ADCC against CMM2 was specific to dPDPN expression, and the Fc region of P38Bf significantly activated canine natural killer (NK) cells in a target antibody-dependent manner. However, ADCC was not induced against KMeC cells, although KMeC cells were weakly recognized by P38Bf. It was reported that the lack of sufficient antigen density on target cells and insufficient antibody–antigen density lead to insufficient ADCC [46,47,48]. Since dPDPN expression on KMeC cells was weaker than that of CMM2, it was considered that dPDPN expression on KMeC cells was not sufficient to induce ADCC by P38Bf and canine natural killer (NK) cells. These results indicated that P38Bf could recognize dPDPN expressed on canine tumor cells at the endogenous expression level and induce ADCC against dPDPN positive canine malignant melanoma.

A preclinical safety trial was performed using only one healthy beagle dog due to animal ethics. Since immunological adverse effects against injected antibodies derived from other species have been reported, chimeric antibodies have been developed and have successfully inhibited immunological adverse effects in human medicine (e.g., Rituximab) [49,50]. Based on these reports, in this study, we used mouse-canine chimeric anti-dPDPN antibody to suppress immunological adverse effects in dogs. Even when injected antibodies were chimeric antibodies, some reports described chimeric antibody-related infusion reactions, which are acute adverse effects and often occur within 30 min to 2 h after administration [49,50,51,52]. Common symptoms and signs are dyspnea, nausea, headache, and abdominal pain [49,50,51,52]. Most reactions were reported to be mild, and it was also reported that only approximately 0.3% of patients have serious infusion reactions with features of anaphylaxis (bronchospasm, hypotension, and angioedema) [49,50,51,52]. In this preclinical safety trial, we carefully evaluated acute adverse effects after four injections of the antibody for 5 h after antibody administration in one experimental beagle dog. There were no symptomatic acute adverse effects or anaphylaxis signs, including abnormality of circulation and respiration (i.e., body temperature, heart rate, respiratory rate, and percutaneous oxygen saturation (SpO_2_)). These findings indicate that the minimum safety of the infusion reaction with P38Bf. Since the rate of infusion-related adverse effects varies among different monoclonal antibodies (rituximab: 77%; trastuzumab: 40%; cetuximab: 15%–20%; bevacizumab: <3%; and panitumumab: 3%) [53], the rate of infusion-related reactions induced by P38Bf administration should be investigated in a large-scale clinical trial.

Considering the chronic adverse effects caused by on-target/off-tumor cytotoxity, no obvious chronic adverse effects were observed by physical examination, blood tests, urinary analysis, computed tomography (CT) analysis, or histopathological analysis in this preclinical safety trial. We hypothesize several mechanisms for the expropriation of this phenomenon. First, P38Bf recognizes dPDPN expressed on tumor cells but not on normal cells. Since PDPN is a glycoprotein that causes tumor-specific aberrant glycosylation in tumor cells [17,31], a cancer-specific anti-PDPN antibody could specifically recognize cancer-specific aberrant glycosylation. P38Bf was designed to recognize only dPDPN expressed on tumor cells but not dPDPN expressed in normal tissues by recognizing cancer-specific aberrant glycosylation [17,36]. In fact, although it has been reported that dPDPN is expressed in various normal tissues, including the kidney, lung, and lymphatic vessels, PMab-38, the original form of P38Bf, did not recognize dPDPN expression in any canine normal tissues [17]. Second, P38Bf might recognize dPDPN expressed on tumor cells and normal cells, but ADCC was only induced against tumor cells that overexpressed dPDPN due to sufficient antibody density. We previously reported that the expression density of dPDPN in tumor cells was stronger than that on normal cells [17]. In addition, it was demonstrated that P38Bf could induce ADCC in dPDPN high-expression canine melanoma cells but not against dPDPN low-expression canine melanoma cells. As another possible mechanism, P38Bf might recognize dPDPN expressed on tumor cells and normal cells, but NK cells were not able to infiltrate into normal tissues and therefore could not be activated there. This hypothesis is supported by many reports that have demonstrated that more NK cells were gathered and activated in canine and human melanoma tissues compared to normal tissues [54]. In this study, we could not clarify the specific mechanism of this phenomenon, and further evaluation is needed.

In the phase I/II clinical trial, three dogs with malignant melanoma were included in this study. Canine melanoma is the most common tumor in canine oral tumors. It accounts for 3% of all neoplasms and up to 7% of all malignant tumors [55,56,57,58]. Generally, complete surgical removal is performed for the management of local tumor control, but almost all dogs with advanced melanoma will progress to lung and lymph node metastasis and death within one year because of resistance to chemotherapy and radiotherapy [55,56,57,58]. A new and effective therapeutic approach for canine malignant melanoma is required. We conducted a phase I/II clinical trial to evaluate the clinical safety and potential efficacy of P38Bf. All treatment-related acute and chronic adverse events were grade 1 or 2 out of 5. Although the acute adverse effect of grade 2 vomiting was observed in one dog, vomiting was transient and manageable with anti-emetic treatment alone. In humans, grade 1–2 nausea and vomiting are common adverse events with antibody-based therapy, occurring in 9.7–33% of patients given trastuzumab, and most events were manageable with symptomatic treatments [59,60,61]. Vomiting induced by P38Bf would also be tolerable and manageable in a clinical setting, and it is considered that vomiting induced by P38Bf is not sufficiently serious to withdraw the clinical trial. In this study, C-reactive protein (CRP), a marker of systemic inflammation, increased mildly in all dogs. CRP is a substance produced by the liver in response to inflammation and is often used as an inflammatory marker [62]. CRP increase was often observed as immune-related adverse events; it has also been reported that approximately 20% of patients show tissue inflammation after rituximab injection [63,64]. This tissue inflammation is called serum sickness, which is a delayed type 3 allergic reaction against chimeric antibodies, and the reaction typically develops one to two weeks after treatment [50,53,64,65]. The increase in CRP in all dogs indicates the occurrence of inflammation due to disease progression, antibody-induced antitumor responses, and/or serum sickness induced by P38Bf injection. We could not completely elucidate the specific cause of CRP increase at this time. In any case, there were no severe treatment-related adverse effects that required withdrawal of the antibody administration in this study, promoting a further investigation of large clinical trials to evaluate the efficacy and safety of P38Bf.

In this study, the responses to P38Bf administration were stable disease (SD) in one dog and progressive disease (PD) in two dogs. Although one dog (#1) with melanoma that was categorized as WHO stage IV had PD, a few metastatic skin lesions ceased rapid growth after the administration of P38Bf. A previous study has reported that PDPN is related to tumor invasion and metastasis, and it was also reported that the PDPN expression density of tumor cell lines derived from metastatic lesions was higher than that of the primary tumor [66]. In this case, it is possible that the difference in responses among metastatic lesions might have caused heterogeneity in PDPN expression density. The clinical response of dog#3, categorized as WHO stage I, was SD. It is well known that the efficacy of cytotoxic antibodies depends on the number of tumor-infiltrating effector cells (e.g., NK cells, macrophages), and cytotoxic antibodies show stronger anti-tumor effects against tumors with a larger number of infiltrating effector cells [67,68]. In canine malignant melanoma, the number of infiltrating effector cells is inversely correlated with tumor stage [69]. It is possible that P38Bf showed slight anti-tumor effects against the dog with stage I malignant melanoma because the efficacy of P38Bf also depends on the number of tumor-infiltrating effector cells. Further investigation into the clinical efficacy of P38Bf in dogs with malignant tumors and the factors that affect the efficacy of P38Bf are needed.

In recent years, new strategies using antibodies, such as antibody–drug conjugate (ADC) and chimeric antigen receptor (CAR) T-cell therapy, have been developed to improve anti-tumor effects. ADC are antibodies connected with anti-tumor cytotoxic molecules that can specifically kill the target cells by a specific linkage, such as trastuzumab-emtansine targeting human epidermal growth factor receptor type2 positive tumors [70,71]. CAR T-cells consist of a single-chain Fab of antibody and T cells and eliminate target cells by cytotoxic T-cell function, such as CD19 CAR-T cell therapy targeting lymphoma and leukemia [72,73,74]. Recently, CAR-T cells targeting PDPN were reported, and anti-PDPN CAR-T cells demonstrated strong anti-tumor effects in orthotopic glioblastomas of the mouse brain [75]. Although P38Bf was safely administered to the four dogs, strong anti-tumor effects by P38Bf were not observed in the phase I/II clinical trial. The anti-PDPN therapy based on ADC and CAR-T strategies would improve the anti-tumor effects of P38Bf.

The limitation of this study was the small sample size. In this study, a total of four dogs were administered P38Bf, but this number was not sufficient for a full evaluation of safety. Antibody concentration was specified as 2.0 mg/kg in this study. Appropriate antibody concentrations should be optimized in further clinical trials. A long-term administration of P38Bf and long-term observation in a large clinical cohort would elucidate the detailed clinical efficacy and safety.

In this study, the efficacy and safety of the cancer-specific anti-canine PDPN chimeric antibody, P38Bf, was evaluated. In the in vitro experiment, our findings suggested that P38Bf has strong ADCC activity and anti-tumor effects against dPDPN expressing canine melanoma cells. A preclinical safety trial and phase I/II clinical trial suggested that P38Bf could be safely administered without severe adverse events, although the observed clinical response was only a potential anti-tumor effect. Further investigation for large clinical trials is needed to clarify the efficacy and safety of P38Bf.

## Figures and Tables

**Figure 1 cells-09-02529-f001:**
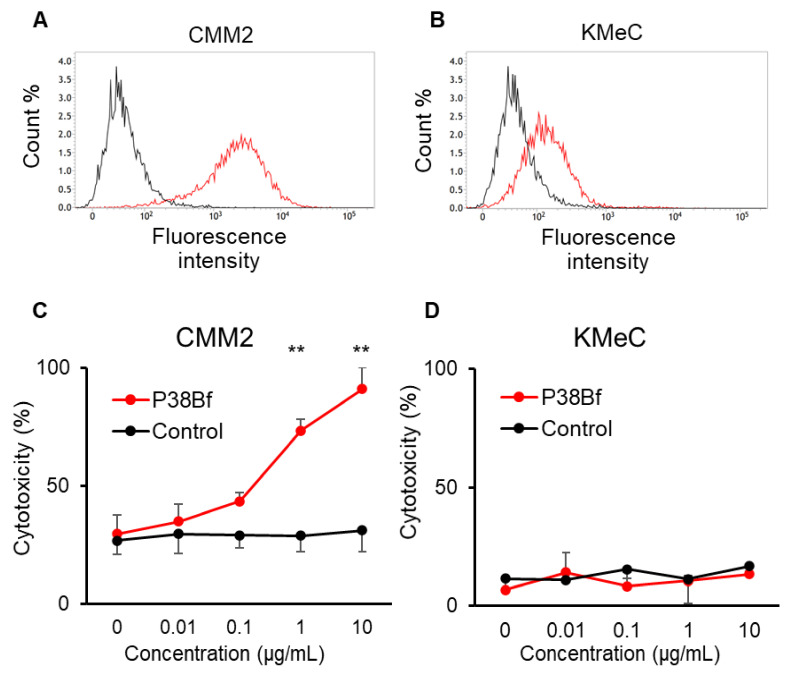
Flow cytometric analysis of CMM2 (**A**) and KMeC (**B**) cells treated with isotype control (black line) and a cancer-specific mouse–dog chimeric anti-PDPN antibody (P38Bf) (red line). P38Bf showed a higher reaction against CMM2 cells compared with KMeC cells. Antibody-Dependent Cellular Cytotoxicity (ADCC) activity of P38Bf against CMM2 (**C**) and KMeC cells (**D**) at various antibody concentrations determined by calcein release assay show significant ADCC activity against CMM2 cells in a dose-dependent manner. Values presented are the mean ± SEM. ** *p* < 0.01.

**Figure 2 cells-09-02529-f002:**
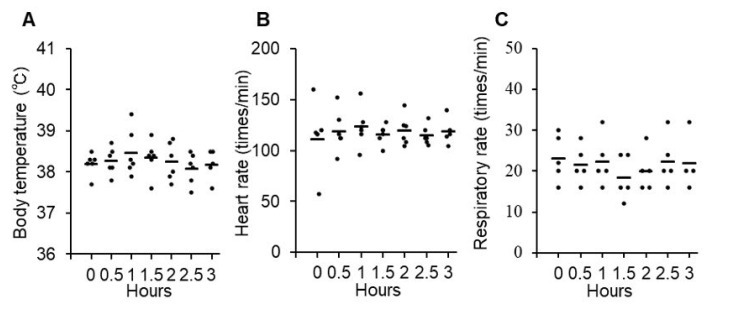
Evaluation of acute adverse effects of P38Bf administration in three dogs. Effects of P38Bf administration on body temperature (**A**), heart rate (**B**) and respiratory rate (**C**). Each dot represents an individual subject at one P38Bf administration and horizontal bars indicate the mean value for all patients at each time point. Significant changes in each parameter were not observed. One-way ANOVA test was performed and p values were 0.77, 0.98, and 0.85, respectively.

**Figure 3 cells-09-02529-f003:**
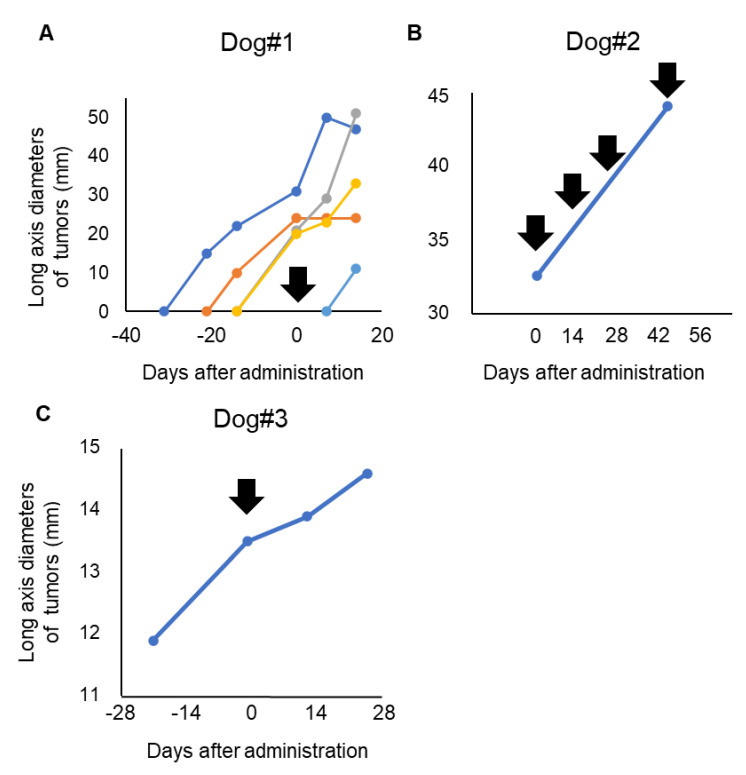
Evaluation of P38Bf efficiency in the clinical trial. Measurement and evaluation of the skin metastasis of Dog #1 (**A**), the oral primary tumor of Dog#2 (**B**), and the oral primary tumor of Dog#3 (**C**). The tumor burden of Dog#1, Dog#2, and Dog#3 was evaluated by measuring the diameter of the long axis. The day of first antibody administration is shown as Day 0, and black arrows indicate the timing of P38Bf administration.

**Table 1 cells-09-02529-t001:** Characteristics of dogs treated with P38Bf.

Case #	Breed	Age (Years)	Sex	Body Weight (kg)	Diagnosis	Primary Site	WHO Stage	Prior Therapy
1	Mix	14.7	Male, castrated	15.5	Malignant melanoma	Tongue	IV	Surgery, radiation, chemotherapy
2	Miniature dachshund	13.7	Female, spayed	4.9	Malignant melanoma	Hard plate	III	Surgery
3	Beagle	14.8	Male	11	Malignant melanoma (amelanotic)	Gingiva	I	Surgery

**Table 2 cells-09-02529-t002:** Treatment-related adverse events of P38Bf administration.

Events	Number of Cases					
	Grade 1	Grade 2	Grade 3	Grade 4	Grade 5	Total
General disorders						
Vomiting	0	1	0	0	0	1
Biochemical parameters						
Increased CRP	3	0	0	0	0	3

**Table 3 cells-09-02529-t003:** Results of dogs treated with P38Bf.

Case #	Dosage	Times Given P38Bf	Treatment Duration	Best overall Response
1	2 mg/kg	1	2 weeks	PD *
2	2 mg/kg	4	9 weeks	PD
3	2 mg/kg	1	2 weeks	SD

* Part of the lesion showed growth inhibition.

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
