# Peer review of "Phase I/II Clinical Trial of the Anti-Podoplanin Monoclonal Antibody Therapy in Dogs with Malignant Melanoma"

_cells, 2020, doi:10.3390/cells9112529_

Round 1

Reviewer 1 Report

Abberantly glycosylated podoplanin is an interesting target for anticancer therapies, especially those basing on cytotoxic abilities of the immune effector cells, e.g. the ADCC-dependent ones or CAR-T cell therapy.

The current manuscript by Kamoto et al. describes the study on safety and efficacy of the recently developed cancer-specific monoclonal anti-canine-podoplanin chimeric antibody (P38Bf) in a I/II phase clinical trial in dogs in a model of canine melanoma.

The study is well designed and clearly presented, both in vitro and in vivo data. The adverse effects observed during the treatment of the dogs were mild and suggest the potential safety of this antibody for the use in subsequent clinical trials. Nevertheless, the antitumor efficacy of the treatment was modest, which perhaps suggests the neccessity for the dose-escalation studies.

Generally, the results presented by the Authors may be further utilized as benchmark settings in preclinical development of anti-podoplanin therapies for the prospective use in humans or directly for the veterinary medicine in other tumor types.

Author Response

Thank you very much. We have made revision according to other reviewers’ comments. We are happy to have your comments again regarding the revised version of the manuscript, if any.

Reviewer 2 Report

The paper titled ‘ Phase I/II Clinical Trial of the Anti-Podoplanin Monoclonal Antibody Therapy in Dogs with Malignant Melanoma’ by Satoshi Kamoto and colleagues reports on a first-in-dog clinical study examining the efficacy and safety of a cancer-specific anti-canine podoplanin (PDPN) chimeric antibody, P38Bf. The authors demonstrate that the target PDPN is present in different dog melanoma cell lines. In vitro efficacy of P38Bf is demonstrated via antibody-mediated cellular toxicity assay using a PDPN high expressing and a PDPN low expressing dog melanoma cell line. Safety of P38Bf is demonstrated in one healthy dog and a phase I/II clinical trial in three dogs with melanoma suggesting that P38Bf could be safely administered without severe adverse events. However, clinical responses include stable disease in one dog and progressive disease in the two remaining dogs.

Although the study shows some encouraging results and might be of interest to readers in the field of cancer research, this study might not be well aligned with the overall scope and aims of the journal Cells, which has a major focus on experimental cytology rather than on clinical and epidemiological studies. In addition, in its current form, the study also contains inaccuracies as outlined in more detail below. Some sections of the paper should be re-written to improve the clarity. My recommendation is therefore to reject this paper.

-Major issues-

INTRODUCTION

Line 61: Please add the name of the anti-PDPN antibody mentioned in ‘As a blocking antibody, the anti-PDPN neutralizing antibody MS-1,…’

Are there any other anti-PDPN antibodies in the literature that have been investigated as anti-cancer therapeutics? If so, why do the authors only report on this MS-1 antibody? This is not clear. Please state if more anti-PDPN antibodies have been investigated in the field of cancer and expand this section of the paper with relevant key antibodies currently under investigation.

FIGURES

Figure 1: Each panel should be annotated. This figure has 4 panels (A, B, C, D), not 2. Please adjust figure and legends as well as any reference to these panels.

Figure legend 1: The authors state that the error bars represent the SEM of three independent experiments, however at the end of the figure legend authors report n = 4? Please correct.

In general, the figure legends should be written concisely: An example is given here for Figure 1 C and D. Replace : ‘ADCC activity of P38Bf against CMM2 and KMeC cells at various antibody concentrations. ADCC activity was determined by calcein release assay. P38Bf demonstrated significant ADCC activity against CMM2 cells in a dose dependent manner Conversely, P38Bf did not exhibit ADCC activity against KMeC cells. All experiments were repeated at least three times. Values presented are the mean ± SEM of three independent experiments. Significantly different from the corresponding control, **p < 0.01. n=4.

By: ‘ ADCC activity of P38Bf against CMM2 (C) and KMeC cells (D) at various antibody concentrations determined by calcein release assay showing significant ADCC activity against CMM2 cells in a dose dependent manner. Values presented are the mean ± SEM. **p < 0.01; n = 4’

Figure 2:

Please label (A, B, C) and define each subpanel in the figure legend of Figure 2.   Please remove full text next to the '0' on the X-axis of second subpanel. Authors covered some text but parts are still visible. 

Supplemental Figure 5A shows CRP results of 2 out of three dogs. Please provide data of all three dogs.

Figure 3: Please include the graphs of tumour lesion measurements for each dog as subpanels (A, dog#1; B, dog#2; C, dog#3). Figure 3 only shows measurement of dog #1.

Figure legend 3 states: ‘Evaluation of the safety and efficacy of P38Bf in clinical trial’. The figure shows tumour measurements of skin metastasis of Dog #1. What part of the graph reports on safety?

RESULTS

Line 294: ‘Three dogs with malignant melanoma that diffusely expressed dPDPN were enrolled in the clinical trial…’ Please included IHC analysis of the dogs that formed basis for enrolment in the study.

-Minor issues-

Line 55: The authors describe ‘Immunotherapy using therapeutic antibodies… has emerged as a new therapeutic strategy for various tumors’.

Instead of ‘has emerged as a new therapeutic strategy’ it would be better to state that ‘therapeutic antibodies have become the predominant class of new drugs developed in recent years.’ The first anti-cancer antibody has been approved by the FDA in 1997. Currently 30 antibodies have been approved for cancer.

Line 55-59: This section could be omitted and replaced by citing recent review papers.

Example: Ruei-Min Lu, Yu-Chyi Hwang, I-Ju Liu, Chi-Chiu Lee, Han-Zen Tsai, Hsin-Jung Li & Han-Chung Wu. Development of therapeutic antibodies for the treatment of diseases. Journal of Biomedical Science volume 27, Article number: 1 (2020)

Line 86: Replace ‘further’ by furthermore’

Line 126: Please write TMB in full before using abbreviation for the first time

Line 178: ‘Computed tomography (CT) was performed to evaluate structural abnormalities in the whole body at the timing of before 7 days and after 56 days of antibody administration. Insert a comma after ‘…at the timing of,’

Line 255: ‘In contrast, there were no significant differences in the ADCC against KMeC cells between P38Bf and the control isotype antibody at any concentration.’

Please rephrase. Example: ‘In contrast, no significant ADCC activity of P38f was measured in KMeC cells (Figure 1D).’

Line 285-288: The authors state: ‘After 56 days of antibody administration, computed tomography (CT) examination was performed to evaluate organic changes by P38Bf between pre-first administration and post-last administration, and there was no detectable change (Supplemental movie 1 (pre-first) and 2 (post-last)).’

It is unclear what expected changes the authors were looking for? I am not convinced this information is relevant or adds value to the importance of this study and could therefore be omitted.

Line 309: Vomiting could be controlled by a single anti-emetic drug (?)… Do the authors mean ‘single anti-emetic drug administration’?

Line 370: ‘In a previous report, the specificity of ADCC activity induced by P38Bf was confirmed, P38Bf showed significant and dose-dependent ADCC activity against CHO/dPDPN cells but not CHO-K1 cells [37].’

Incorrect reference is provided, reference [37] should be replaced by [36]. Please rephrase this sentence for clarity. Example: ‘The specificity of ADCC activity induced by P38Bf was originally demonstrated using CHO/dPDPN cells (dog PDPN positive) and CHO-K1 cells (dog PDNP negative cells) [36], and therefore these cell lines were included in the current study as controls.’

Line 376: ‘It was reported that the lack of sufficient antigen density on target cells and insufficient antibody-antigen density lead to insufficient ADCC [46–48]’. Please rephrase, example: ‘This aligns with previous reports showing that insufficient PDPN antigen density leads to insufficient ADCC [46–48].’

Line 381: …expression level and induced ADCC against dPDPN positive canine malignant melanoma. Replace ‘induced’ by ‘induce’

Line 382: ‘Because immunological adverse effects against injected antibodies derived from other species have … immunological adverse effects in human medicine (e.g., Rituximab).’ Provide a reference.

Round 2

Reviewer 2 Report

The paper titled ‘ Phase I/II Clinical Trial of the Anti-Podoplanin Monoclonal Antibody Therapy in Dogs with Malignant Melanoma’ has been revised by Satoshi Kamoto and colleagues.

In my original review I did not realize this paper was submitted towards publication in a special edition of the journal ‘Cells’ covering "Structure and Function of Podplanin in Disease". I accept the explanation offered by the corresponding author and agree that their paper is within the scope of this special issue.

In addition, the authors have significantly improved the quality of the paper including the addition of new data, updating of figures and legends, rewriting and including appropriate references. Therefore, my recommendation is to accept this manuscript for publication in the special issue of the journal of Cells once the authors have been able to adjust some minor issues:

Minor issues:

  • Figure legend 3 and figure 3: Remove ‘The’ in title of each subpanel e.g. The dog#1 --> Dog#1

Make corrections in Figure legend 3 as follows:

Evaluation of P38Bf efficiency in the clinical trial. Measurement and evaluation of the skin metastasis of Dog #1 (A), the oral primary tumor of Dog#2 (B), and the oral primary tumor of Dog#3 (C). The tumor burden of Dog#1, Dog#2 and Dog#3 was evaluated by measuring the diameter of the long axis. The day of first antibody administration is shown as Day0, and black arrows indicate the timing of P38Bf administration.

  • Supplementary legend Fig5. and figure: Remove ‘The’ in title of each subpanel: e.g. The dog#1--> Dog#1. Please also make changes to the legend.

  • Change Table legend of Table 1 as follows:

Supplementary Table 1: Detailed results of the blood test of one healthy dog.
